# Graph Alignment via Birkhoff Relaxation

**Sushil Mahavir Varma**
Industrial and Systems Engineering
University of Michigan Ann Arbor
Michigan, USA
sushilv@umich.edu

**Irène Waldspurger**
CNRS, INRIA
Université Paris Dauphine
Paris, France
waldspurger@ceremade.dauphine.fr

**Laurent Massoulié**
INRIA, DI/ENS
PSL Research University
Paris, France
laurent.massoulie@inria.fr

## Abstract

We consider the graph alignment problem, wherein the objective is to find a vertex correspondence between two graphs that maximizes the edge overlap. The graph alignment problem is an instance of the quadratic assignment problem (QAP), known to be NP-hard in the worst case even to approximately solve. In this paper, we analyze Birkhoff relaxation, a tight convex relaxation of QAP, and present theoretical guarantees on its performance when the inputs follow the Gaussian Wigner Model. More specifically, the weighted adjacency matrices are correlated Gaussian Orthogonal Ensemble with correlation $1/\sqrt{1+\sigma^2}$. Denote the optimal solutions of the QAP and Birkhoff relaxation by $\Pi^\star$ and $X^\star$ respectively. We show that $\|X^\star - \Pi^\star\|_F^2 = o(n)$ when $\sigma = o(n^{-1})$ and $\|X^\star - \Pi^\star\|_F^2 = \Omega(n)$ when $\sigma = \Omega(n^{-0.5})$. Thus, the optimal solution $X^\star$ transitions from a small perturbation of $\Pi^\star$ for small $\sigma$ to being well separated from $\Pi^\star$ as $\sigma$ becomes larger than $n^{-0.5}$. This result allows us to guarantee that simple rounding procedures on $X^\star$ align $1 - o(1)$ fraction of vertices correctly whenever $\sigma = o(n^{-1})$. This condition on $\sigma$ to ensure the success of the Birkhoff relaxation is state-of-the-art.

## 1 Introduction

Consider two undirected graphs $G_1$ and $G_2$ with vertices $[n]$. The *graph matching/alignment* problem is defined as finding a mapping between the vertices of $G_1$ and $G_2$ such that the edge overlap is maximized. The graph isomorphism problem is a special case when $G_1 = G_2$ up to the permutation of the vertices. The graph matching problem has widespread applications in network de-anonymization [28], computational biology [30], pattern recognition [6], etc., underlining the need to design and study efficient algorithms for this problem.

More formally, let $A$ and $B$ be the adjacency matrices (possibly weighted) of $G_1$ and $G_2$ respectively, then, the vertex correspondence that maximizes the edge overlap is given by the optimal solution of the following quadratic assignment problem:

$$\Pi^\star = \arg \min_{X \in \mathcal{P}_n} \|AX - XB\|_F^2, \tag{1}$$

where $\mathcal{P}_n$ is the set of all $n \times n$ permutation matrices. Quadratic assignment problems are NP-hard in the worst case and are known to be difficult to even solve approximately [24]. However, for typical graphs, one could expect to efficiently solve the graph matching problem. Indeed,

polynomial time algorithms are known for the graph isomorphism problem when $G_1 = G_2$ is an Erdős-Rényi random graph [5, 7]. More generally, the setting with noise, where $G_1, G_2$ are not exactly equal but are *correlated* Erdős-Rényi graphs, has been extensively studied in the last decade [8, 9, 11, 12, 18, 17, 26, 25, 27, 2].

A popular class of algorithms in practice with good empirical and computational performance is the convex relaxations of (1). We are interested in the Birkhoff relaxation, which is a tight convex relaxation, wherein the set of permutation matrices $\mathcal{P}_n$ is replaced by its convex hull to get

$$X^\star = \arg \min_{X \in \mathcal{B}_n} \|AX - XB\|_F^2, \tag{2}$$

where $\mathcal{B}_n$ is the Birkhoff polytope, i.e., the set of all doubly stochastic matrices. Such a relaxation has been an attractive approach in practice, with strong empirical performance for shape matching [1] and ordering images in a grid [10]. It is also observed in [23] that Birkhoff relaxation, when combined with indefinite relaxation, yields excellent empirical performance on real datasets. More generally, similar relaxations are proposed for gene sequencing [14], and computer vision [4].

While these results highlight the usefulness of the Birkhoff relaxation to practice due to its attractive computational and empirical performance, there is limited understanding of its theoretical performance. Theoretical guarantees for alternate convex relaxations of the QAP (1) are available in the literature [11, 12, 2], however, they lack in their empirical performance compared to the Birkhoff relaxation (see Section 5). Motivated by this gap in the literature, we provide the first theoretical guarantees on the performance of the Birkhoff relaxation (2) when $A$, $B$ are sampled from the Gaussian Wigner Model with correlation $1/\sqrt{1 + \sigma^2}$. Imposing such distributional assumptions on the input graphs allows for mathematical analysis, as otherwise, graph matching is an NP-hard problem in the worst case.

## 1.1 Gaussian Wigner Model

We say that a matrix $A := \{A_{ij}\}_{i,j \in [n]}$ is a (GOE) matrix if $A_{ii} \sim N(0, 2/n)$ for all $i \in [n]$, $A_{ij} = A_{ji} \sim N(0, 1/n)$ for all $i \neq j$, and $\{A_{ij} : i \leq j\}$ are mutually independent. We say that $A, B \in \mathbb{R}^{n \times n}$ follows the Gaussian Wigner Model when for all $i, j \in [n]$, we have $B_{\pi^\star(i), \pi^\star(j)} = A_{ij} + \sigma Z_{ij}$, where $A, Z$ are i.id. GOE matrices Alternatively, we can write $B = (\tilde{\Pi}^\star)^T (A + \sigma Z)(\tilde{\Pi}^\star)$, where $\tilde{\Pi}^\star_{ij} = \mathbb{1}\{j = \pi^\star(i)\}$. Given an observation of $A$ and $B$, the goal is to infer the ground truth $\pi^\star$ that aligns $A$ and $B$, which corresponds to the optimal solution of QAP (1) for $\sigma^2 \leq O(n/\log n)$ by [15, 32]. In particular, [15, 32] shows that $\tilde{\Pi}^\star = \Pi^\star$ with probability $1 - o(1)$ for $\sigma^2 \leq O(n/\log n)$ and $n$ large enough. We fix $\sigma \leq 1$ for simplicity.

## 1.2 Main Contributions

In this paper, we establish bounds on the distance of the optimal solution of the Birkhoff relaxation $X^\star$ from the optimal solution of the QAP $\Pi^\star$, which is equal to the true permutation $\tilde{\Pi}^\star$ with high probability. Our contributions are two-fold.

Firstly, we show that $\|X^\star - \Pi^\star\|_F^2 = o(n)$ whenever $\sigma = o(n^{-1})$, asserting that $X^\star$ is close to the true permutation $\Pi^\star$. This bound in turn implies that one can use simple rounding procedures on $X^\star$ to correctly align $1 - o(1)$ fraction of vertices. Our proof technique is motivated by the literature on the sensitivity analysis of convex problems. In particular, we formulate the dual of a version of (2) for $\sigma = 0$. The novelty of the proof lies in the careful construction of a feasible dual solution, which in turn, provides us useful bounds on $X^\star$. Secondly, we show that if $\sigma = \Omega(n^{-0.5})$, then $\|X^\star - \Pi^\star\|_F^2 = \Omega(n)$, and so, $X^\star$ is far away from the true permutation $\Pi^\star$.

The above two results establish a phase transition in the behavior of (2): the optimal solution $X^\star$ transitions from being a small perturbation of $\Pi^\star$ for slowly growing $\sigma$ to moving far away from $\Pi^\star$ as $\sigma$ becomes greater than $1/\sqrt{n}$. In the next section, we discuss that the population version of (2) exhibits such a phase transition at $1/\sqrt{n}$ and so we believe our sufficient condition on $\sigma$ is loose, i.e., one can possibly show $\|X^\star - \Pi^\star\|_F^2 = o(n)$ for $\sigma = o(n^{-0.5})$. Nonetheless, to the best of our knowledge, the sufficient condition $\sigma = o(n^{-1})$ is the state-of-the-art to ensure that (2) succeeds in recovering the true permutation.

Note that $\sigma = 0$ is the special case of the graph isomorphism problem. As the value of $\sigma$ increases, the correlation between the two graphs reduces, which makes it harder to align them. In other words, noise increases, making it harder to extract the correct signal. Thus, $\sigma$ serves as a tuning parameter to increase the hardness of the problem. The takeaway is then to establish that the Birkhoff relaxation succeeds for large values of $\sigma$, establishing the robustness of the relaxation. We make progress in this direction by improving upon the previously known result [2] that shows the simplex relaxation succeeds for $\sigma = 0$. These results contribute to explaining the strong empirical performance of the Birkhoff relaxation.

### 1.3 Related Work

Regarding practical algorithms for graph alignment, there is a rich literature on optimization-based methods [21, 10, 22, 11, 2] and simple spectral-based methods [31, 13, 16]. The works [11, 2] focus on the Gaussian Wigner model and are closest to our paper. In particular, [11] proposes the GRAMPA algorithm which solves the following convex relaxation of (1):

$$\arg \min_{X:\mathbf{1}^T X\mathbf{1}=n} \|AX - XB\|_F^2 + \eta\|X\|_F^2, \quad \text{for some } \eta > 0. \tag{3}$$

Note that the optimization problem above further relaxes (2) and adds a quadratic regularizer. The authors in [11] establish that GRAMPA (3) succeeds in recovering $\Pi^\star$ whenever $\sigma = O(1/\log n)$. While [11] asserts that GRAMPA succeeds with a weaker condition on $\sigma$ than ours, its empirical performance is observed to be worse than the Birkhoff relaxation (2). In addition, GRAMPA requires one to tune the regularization parameter $\eta$ which is completely avoided for the Birkhoff relaxation (2). In addition, [2] analyzes the Simplex Relaxation defined as follows:

$$\arg \min_{X:\mathbf{1}^T X\mathbf{1}=n, X\geq 0} \|AX - XB\|_F^2. \tag{4}$$

They prove that the simplex relaxation (4) succeeds for $\sigma = 0$. On the other hand, we consider a tighter convex relaxation and allow $\sigma$ to be non-zero. Comparison to these methods is summarized in Table 1. We refer the reader to [19, 17, 9, 27] and the references within for analysis beyond the GOE setting.

| Paper | Algorithm Type | Algorithm Name | Noise Threshold |
|---|---|---|---|
| [16] | Top Eigenvector Alignment | EIG1 | $\sigma = \Theta(n^{-7/6})$ |
| [11] | Regularized Convex Relaxation | GRAMPA (3) | $\sigma = O(1/\log n)$ |
| [2] | Simplex Relaxation | Simplex (4) | $\sigma = 0$ |
| Our work | Birkhoff Relaxation | Birkhoff (2) | $\sigma = O(n^{-1})$ |

Table 1: Comparison of the spectral and optimization-based algorithms for graph alignment on the Gaussian Wigner Model.

## 2 Main Result

We now present the main theorem of the paper that characterizes when $X^\star$ is well separated from $\Pi^\star$ and when $X^\star$ is a small perturbation of $\Pi^\star$.

**Theorem 1.** *Let $A, B \in \mathbb{R}^{n \times n}$ follow the Gaussian Wigner model with some $\sigma \leq 1$ (possibly dependent on $n$) and let $\Pi^\star$ and $X^\star$ be defined as in (1) and (2) respectively. Then, for any fixed $\epsilon > 0$ and $n$ large enough, the following statements hold with probability $1 - o(1)$.*

- *Well-Separation: When $\sigma \geq n^{-0.5+\epsilon}$ then $\|X^\star - \Pi^\star\|_F^2 \geq \delta n$ for some $\delta > 0$.*

- *Small-Perturbation: When $\sigma \leq n^{-1-\epsilon}$ then $\|X^\star - \Pi^\star\|_F^2 \leq 10n^{1-\epsilon/32}$.*

Using the small-perturbation result above, one can implement a simple rounding procedure on $X^\star$ to recover a $1 - o(1)$ fraction of the permutation $\pi^\star$ correctly.

**Corollary 2.** *Under the hypothesis of Theorem 1, let $\hat{\pi}(i) = \arg\max_j X_{ij}^\star$ for all $i \in [n]$. If $\sigma \leq n^{-1-\epsilon}$ for some $\epsilon > 0$, then $\sum_{i=1}^n \mathbb{1}\{\hat{\pi}(i) \neq \pi^\star(i)\} = o(n)$ with probability $1 - o(1)$ for $n$ large enough.*

In practice, one may want to implement more sophisticated rounding procedure such as Hungarian projection. Theorem 1 ensures that Hungarian projection also succeeds to recover a $1 - o(1)$ fraction of the permutation $\pi^\star$ correctly.

**Corollary 3.** *Under the hypothesis of Theorem 1, let $\tilde{\Pi} \in \arg\max_{\Pi \in \mathcal{P}_n} \langle X^\star, \Pi \rangle$. If $\sigma \leq n^{-1-\epsilon}$ for some $\epsilon > 0$, then $\langle \tilde{\Pi}^\star, \tilde{\Pi} \rangle \geq n - o(n)$ with probability $1 - o(1)$ for $n$ large enough.*

We refer the reader to Appendix A for proof details. Note that a similar negative result for the well-separated case may not hold. In particular, while $X^\star$ is far away from $\Pi^\star$ for $\sigma = \Omega(n^{-0.5})$, it does not imply that the Birkhoff relaxation fails to recover the permutation $\pi^\star$ correctly, as rounding procedures like above can be used to post-process $X^\star$ which could recover $\Pi^\star$. As our proof technique is based on sensitivity analysis of convex problems, we do not explore this direction in this paper, except in Section 5, where we conduct numerical experiments to provide insights.

To gain intuition, consider the population version of (2), where we replace the objective with its expected value to get the following optimization problem for $\tilde{\Pi}^\star = I$:

$$\min_{X \in \mathcal{B}_n} (2 + \sigma^2)(n+1)\|X\|_F^2 - 2\text{Tr}(X)^2 - 2\langle X, X^T \rangle,$$

whose optimal value is given by

$$\bar{X}^\star = \epsilon I + \frac{1 - \epsilon}{n} J, \quad \text{where} \quad \epsilon = \frac{2}{2 + \sigma^2(n+1)},$$

where we denote the all-ones $n \times n$ matrix by $J$. The above can be verified using the KKT conditions. Now, using the above characterization, we get

$$\|I - \bar{X}^\star\|_F = (1 - \epsilon)\left\|I - \frac{J}{n}\right\| = (1 - \epsilon)\sqrt{n-1} \approx \frac{\sigma^2 n\sqrt{n}}{2 + \sigma^2 n} = \begin{cases} \Theta(\sqrt{n}) & \text{if } \sigma\sqrt{n} = \Omega(1) \\ o(\sqrt{n}) & \text{if } \sigma\sqrt{n} = o(1). \end{cases}$$

The first case above asserts that whenever $\sigma \gg n^{-0.5}$, we have $\|I - \bar{X}^\star\| = \Theta(\sqrt{n})$, i.e., $\bar{X}^\star$ is far from $I$, so, we should not expect $X^\star$ to be close to $I$ as well. The first assertion of Theorem 1 formalizes this intuition. The above equation also shows that $\bar{X}^\star$ is close to $I$ whenever $\sigma \ll n^{-0.5}$, so we expect $X^\star$ to be a small perturbation of $I$. Theorem 1 formalizes this only for $\sigma \ll n^{-1}$, which only partially resolves this case. However, analyzing the Birkhoff convex relaxation (2) is known to be a challenging task, and this paper provides the first results in understanding this relaxation. In particular, one of the main difficulties is in handling the non-negativity constraints. The paper [11] circumvents such a difficulty by relaxing the non-negativity constraints and compensating for it with a quadratic regularizer in the objective function. Such a modification allows them to obtain a closed-form expression of $X^\star$, which is then shown to satisfy certain desirable properties (diagonal dominance). More recently, [2] preserves the non-negativity constraints, but the result is restricted to $\sigma = 0$ and the proof exploits the structural properties of (2) that hold only when $B = A$. We, on the other hand, tackle the challenges head-on presented by non-negativity constraints and also allow $\sigma > 0$. Our proof technique is to carefully construct a suitable feasible solution of the dual of (2), which provides required bounds on $X^\star$. The main difficulty in constructing such a feasible dual certificate is its high dimension as the non-negativity constraints result in $n^2$ dual variables.

## 3 Proof of Theorem 1

Without loss of generality, we assume that $\tilde{\Pi}^\star = I$. Indeed, if $\tilde{\Pi}^\star \neq I$, then, we have

$$\|AX - XB\|_F = \|AX - X(\tilde{\Pi}^\star)^T(A + \sigma Z)\tilde{\Pi}^\star\|_F = \|AX(\tilde{\Pi}^\star)^T - X(\tilde{\Pi}^\star)^T(A + \sigma Z)\|_F.$$

Thus, defining

$$\tilde{X} = \arg\min_{X \in \mathcal{B}_n} \|AX - X(A + \sigma Z)\|_F^2,$$

we conclude that $\tilde{X} = X^\star(\tilde{\Pi}^\star)^T$ and so $\|X^\star - \Pi^\star\|_F = \|\tilde{X} - \Pi^\star(\tilde{\Pi}^\star)^T\|_F = \|\tilde{X} - I\|_F$, where the last equality holds as $\tilde{\Pi}^\star = \Pi^\star$ with probability $1 - o(1)$ by [15, 32]. So, in summary, we have $\tilde{\Pi}^\star = \Pi^\star = I$ without loss of generality.

## 3.1 Part I: Well-Separation

*Proof of Theorem 1 (Part I):.* We start by upper bounding $\|AX^\star - X^\star B\|$ by the objective function value of (2) for $J/n$.

**Claim 4.** *For $n$ large enough, with probability at least $1 - 4ne^{-n^\epsilon/4}$, we have*

$$\|AX^\star - X^\star B\|_F^2 \leq \frac{1}{n^2}\|AJ - JB\|_F^2 \leq 9n^\epsilon$$

The proof details of the above claim is deferred to Appendix B.1. We now construct a lower bound on $\|AX^\star - X^\star B\|_F^2$ as a function of $\|X^\star - I\|_F$. We have

$$
\begin{aligned}
\|AX^\star - X^\star B\|_F^2 &= \|A(I - X^\star) - (I - X^\star)B + \sigma Z\|_F^2 \\
&= \|A(I - X^\star) - (I - X^\star)B\|_F^2 + \sigma^2\|Z\|_F^2 + 2\sigma\langle A(I - X^\star) - (I - X^\star)B, Z\rangle \\
&= \|A(I - X^\star) - (I - X^\star)B\|_F^2 + \sigma^2\|Z\|_F^2 + 2\sigma\langle AZ - ZB, I - X^\star\rangle \\
&= \|A(I - X^\star) - (I - X^\star)B\|_F^2 + \sigma^2\|Z\|_F^2 - 2\sigma\langle AZ - ZA, X^\star\rangle - 2\sigma^2\langle Z^2, I - X^\star\rangle \\
&\geq \frac{\sigma^2 n}{2} - 2\sigma\max_{i\neq j}\left|(AZ - ZA)_{ij}\right|\sum_{i\neq j}X_{ij}^\star - 2\sigma^2\|Z^2\|_F\|I - X^\star\|_F, \quad (5)
\end{aligned}
$$

where the last inequality is true because $\|Z\|_F^2 = \frac{2}{n}\|\tilde{z}\|_2^2$, where $\tilde{z} \sim N(0, I_{n(n+1)/2})$. Thus, by [11, Lemma 15], with probability at least $1 - e^{-\sqrt{n}}$, we have $\|Z\|_F^2 \geq n - \tilde{c}\sqrt{n} \geq n/2$, for a sufficiently large $n$ since $\tilde{c} > 0$ is a constant independent of $n$. We now bound $\max_{i\neq j}\left|(AZ - ZA)_{ij}\right|$ in the following claim.

**Claim 5.** *For $n$ large enough, w.p. $1 - 2n^2 e^{-n^\epsilon}$, we have $\max_{i\neq j}\left|(AZ - ZA)_{ij}\right| \leq 8n^{\epsilon/2 - 0.5}$.*

The proof details of the claim is deferred to Appendix B.2. Substituting the above bound back in (5) and noting that $\sum_{i\neq j}X_{ij}^\star \leq n$ as $X^\star \in \mathcal{B}_n$, we get

$$
\begin{aligned}
\|AX^\star - X^\star B\|_F^2 &\geq \frac{\sigma^2 n}{2} - 16\sigma n^{1/2 + \epsilon/2} - 2\sigma^2\|Z^2\|_F\|I - X^\star\|_F \\
&\geq \frac{\sigma^2 n}{4} - 2c\sigma^2\sqrt{n}\|I - X^\star\|_F. \quad (6)
\end{aligned}
$$

where the last inequality holds for $n$ large enough as $\sigma^2 n = \Theta(n^{2\epsilon})$ and $\sigma n^{1/2 + \epsilon/2} = \Theta(n^{3\epsilon/2})$. In addition, we use the inequality $\|Z^2\|_F \leq \sqrt{n}\|Z^2\|_2 \leq \sqrt{n}\|Z\|_2^2 \leq c\sqrt{n}$ for some $c > 0$ sufficiently large, which holds with probability at least $1 - e^{-n}$ by [3, Lemma 6.3]. Now combining the upper bound in Claim 4 with the lower bound in (6), we get

$$2c\sigma^2\sqrt{n}\|I - X^\star\|_F \geq \frac{\sigma^2 n}{4} - 9n^\epsilon \overset{*}{\geq} \frac{\sigma^2 n}{8} \implies \|I - X^\star\|_F \geq \frac{\sqrt{n}}{16c}.$$

where $(*)$ holds for $n$ large enough. The above assertion holds with probability at least $1 - 5n^2 e^{-n^\epsilon/4} \geq 1 - e^{-n^{\epsilon/2}}$ for $n$ large enough by the union bound over all the high probability bounds in the proof. This completes the proof with $\delta = 1/(16c)$, a universal constant. $\quad\square$

Note that the above proof does not rely on specific properties of a GOE matrices. Thus, such a conclusion would hold for more general symmetric matrices with i.id. entries that concentrate sufficiently, e.g., subgaussian concentration.

## 3.2 Part II: Small-Perturbation

The main idea of the proof is to construct a suitable dual feasible solution that provides an upper bound on the off-diagonal entries of $X^\star$. As the primal problem (2) is non-linear, resulting in an involved dual problem, we start up considering a simpler optimization problem corresponding to $\sigma = 0$ in (2). As we will see below, this analysis, in turn, allows us to conclude meaningful bounds on $X^\star$ for $\sigma = o(n^{-1})$. For $\sigma = 0$, we have $\min_{X \in \mathcal{B}_n}\|AX - XA\|_F^2$. Note that $X = I$ is an

optimal solution as it results in 0 objective function value. Thus, any optimal solution must satisfy $AX - XA = 0$, and so, we can further simplify to write the following optimization problem.

$$\min_{X \in \mathcal{B}_n} 0 \quad \text{subject to } AX - XA = 0.$$

We introduce dual variables $R \in \mathbb{R}_+^{n \times n}, \mu, \tilde{\mu} \in \mathbb{R}^n$ for the non-negativity, row sum, and column sum constraints, and $M \in \mathbb{R}^{n \times n}$ for the constraint $AX - XA = 0$. Then, the Lagrangian is given by

$$\min_X \max_{R \geq 0, \mu, \tilde{\mu}, M} \langle AX - XA, M \rangle - \langle R, X \rangle + \mu^T X \mathbf{1} + \mathbf{1}^T X \tilde{\mu} - \mu^T \mathbf{1} - \tilde{\mu}^T \mathbf{1}.$$

Now, we swap the order of the min and the max to obtain the dual problem.

$$\max_{R \geq 0, \mu, \tilde{\mu}, M} \min_X \langle AX - XA, M \rangle - \langle R, X \rangle + \mu^T X \mathbf{1} + \mathbf{1}^T X \tilde{\mu} - \mu^T \mathbf{1} - \tilde{\mu}^T \mathbf{1}$$

$$= \max_{R \geq 0, \mu, \tilde{\mu}, M} \min_X \langle AM - MA - R + \mu \mathbf{1}^T + \mathbf{1} \tilde{\mu}^T, X \rangle - \mu^T \mathbf{1} - \tilde{\mu}^T \mathbf{1}$$

$$= \max_{R \geq 0, \mu, \tilde{\mu}, M} -\mu^T \mathbf{1} - \tilde{\mu}^T \mathbf{1} \quad \text{subject to, } AM - MA - R + \mu \mathbf{1}^T + \mathbf{1} \tilde{\mu}^T = 0.$$

We are now looking for a dual feasible solution $(R, \mu, \tilde{\mu}, M)$ that also satisfies strong duality. We construct an approximately feasible dual solution, i.e., we have

$$\mu^T \mathbf{1} + \tilde{\mu}^T \mathbf{1} = 0, \; R \geq 0, \; AM - MA - R + \mu \mathbf{1}^T + \mathbf{1} \tilde{\mu}^T \approx 0 \tag{7}$$

which implies for any $X \in \mathcal{B}_n$ with $AX - XA = 0$

$$\langle R, X \rangle = \langle R, X \rangle - \langle AX - XA, M \rangle - \mu^T \mathbf{1} - \tilde{\mu}^T \mathbf{1}$$

$$= \langle R, X \rangle - \langle AM - MA, X \rangle - \langle \mu \mathbf{1}^T + \mathbf{1} \tilde{\mu}^T, X \rangle$$

$$= \langle R - AM + MA - \mu \mathbf{1}^T - \mathbf{1} \tilde{\mu}^T, X \rangle \approx 0.$$

To get a meaningful bound, we set $R = J - I$ and appropriately construct $(M, \mu, \tilde{\mu})$ which implies

$$\sum_{j \neq i} X_{ij} \approx 0.$$

We formalize this argument in the lemma below:

**Lemma 6.** *Under the hypothesis of Theorem 1, for any $X \in \mathcal{B}_n$ and any $\epsilon > 0$, for $n$ large enough, we have*

$$\sum_{j \neq i} X_{ij} \leq 2n^{3/2 + 7\epsilon/8} \|AX - XA\|_F + 4n^{1-\epsilon/32} \quad \text{w.p. } 1 - o_n(1).$$

Now to show that the bound obtained in the above lemma is small enough, we upper bound $\|AX - XA\|_F$ for the optimal solution $X = X^\star$ in the following lemma.

**Lemma 7.** *Under the setting of Theorem 1, there exists a constant $c > 0$ such that $\|AX^\star - X^\star A\|_F \leq c\sigma\sqrt{n}$ with probability at least $1 - e^{-n}$ for $n$ large enough.*

The proof strategy of the above lemma is as follows: As $I \in \mathcal{B}_n$ is a feasible solution to (2), we immediately obtain $\|AX^\star - X^\star B\|_F \leq \|A - B\|_F = \sigma\|Z\|_F = O(\sigma\sqrt{n})$, where the last inequality follows as $\|Z\|_F = O(\sqrt{n})$ for a GOE matrix. The rest of the argument is to show that $\|AX^\star - X^\star A\|_F \lesssim \|AX^\star - X^\star B\|_F + O(\sigma\sqrt{n})$ by writing $B = A + \sigma Z$, expanding $\|AX^\star - X^\star B\|_F$, and appropriately bounding terms involving $\sigma$. The proof details are deferred to Appendix C.1.

*Proof of Theorem 1 (Part II):.* By Lemma 6 and Lemma 7, for $\sigma = n^{-1-\epsilon}$ we get

$$\sum_{i,j \in [n]: i \neq j} X_{ij}^\star \leq 2c\sigma n^{2+7\epsilon/8} + 4n^{1-\epsilon/32} = 2cn^{1-\epsilon/8} + 4n^{1-\epsilon/32} \leq 5n^{1-\epsilon/32}, \tag{8}$$

where the first inequality holds with probability at least $1 - e^{-n} - o_n(1) = 1 - o_n(1)$ by the union bound. The last inequality holds for all $n$ large enough (depending on $\epsilon$). We are now ready to obtain a bound on $\|X^\star - I\|_F$.

$$\|X^\star - I\|_F^2 = \sum_{i,j\in[n]:i\neq j}(X_{ij}^\star)^2 + \sum_{i=1}^n(1-X_{ii}^\star)^2 = \|X^\star\|_F^2 + n - 2\sum_{i=1}^n X_{ii}^\star \overset{*}{\leq} 2n - 2\sum_{i=1}^n X_{ii}^\star$$

$$= 2\sum_{i,j\in[n]:i\neq j}X_{ij}^\star \overset{(8)}{\leq} 10n^{1-\epsilon/32},$$

where $(*)$ follows as $\|X\|_F^2 \leq n\max_{j\in[n]}\sum_{i=1}^n X_{ij}^2 \leq n\max_{j\in[n]}\sum_{i=1}^n |X_{ij}| = n$ for any $X \in \mathcal{B}_n$. This completes the second part of the proof of Theorem 1.

**Remark 8** (Generalization beyond GOE Matrices). *The main difficulties are to get a handle on the eigenvalues and eigenvectors of $A$. In particular, we explicitly use the properties of a GOE matrix in two places in the proof. (1) Eigenvalue separation [Claim 8]: We use tail bounds on the eigenvalues of a random matrix from [5]. The results of [5] are applicable for all Wigner matrices (i.id. subgaussian entries), and so Claim 8 can be extended to Wigner matrices. (2) Eigenvector Concentration [Claim 7]: To prove Claim 7, we rely on the fact that the orthonormal eigenvectors of a GOE matrix is uniformly distributed on $\mathcal{S}_{n-1}$ to obtain upper and lower concentrations on $\langle\mathbf{1}, u_i\rangle$. This step is the bottleneck, as we require such concentration results for general Wigner matrices.*

## 4 Proof of Lemma 6

*Proof of Lemma 6.* For the GOE matrix $A$, let $\{\lambda_i\}_{i=1}^n$ be the set of eigenvalues and $\{u_i\}_{i=1}^n$ be the corresponding set of orthonormal eigenvectors. Now, we construct a dual feasible solution $(R, M, \mu, \tilde{\mu})$ using eigenvectors of $A$ as the basis vectors. First, we set $\tilde{\mu} = \mathbf{0}$ and

$$R = \sum_{i,j=1}^n w_{ij}u_iu_j^T = J - I, \quad \text{with } w_{ij} = \langle u_i, \mathbf{1}\rangle\langle u_j, \mathbf{1}\rangle - \mathbb{1}\{i = j\}.$$

Now, to ensure approximate dual feasibility and strong duality as in (7), we carefully set $(M, \mu)$ as follows. We have

$$\mu = \sum_{i=1}^n w_iu_i \quad \text{with } w_i = \langle u_i, \mathbf{1}\rangle + \frac{C-1}{\langle u_i, \mathbf{1}\rangle}\mathbb{1}\left\{|\langle u_i, \mathbf{1}\rangle| \geq n^{-\epsilon/16}\right\}$$

where $C = 1 - \frac{n}{\#\left\{|\langle u_i, \mathbf{1}\rangle| \geq n^{-\epsilon/16}\right\}}$, with $\#\left\{|\langle u_i, \mathbf{1}\rangle| \geq n^{-\epsilon/16}\right\} = \sum_{i=1}^n \mathbb{1}\left\{|\langle u_i, \mathbf{1}\rangle| \geq n^{-\epsilon/16}\right\}$. Note that $\langle\mu, \mathbf{1}\rangle = 0$ which ensures strong duality. One can quickly verify it as follows:

$$\langle\mu, \mathbf{1}\rangle = \sum_{i=1}^n w_i\langle u_i, \mathbf{1}\rangle = \sum_{i=1}^n \langle u_i, \mathbf{1}\rangle^2 + (C-1)\#\left\{|\langle u_i, \mathbf{1}\rangle| \geq n^{-\epsilon/16}\right\} = \sum_{i=1}^n \langle u_i, \mathbf{1}\rangle^2 - n = 0.$$

Next, we set

$$M = \sum_{i,j\in[n]:i\neq j}\frac{\tilde{w}_{ij}}{\lambda_i - \lambda_j}u_iu_j^T \quad \text{with } \tilde{w}_{ij} = (\langle u_i, \mathbf{1}\rangle\langle u_j, \mathbf{1}\rangle - \langle u_i, \mathbf{1}\rangle w_j)\mathbb{1}\{i \neq j\}.$$

Now, we show that $(R, \mu, \tilde{\mu}, M)$ is approximately dual feasible as in (7). By construction, we have

$$R - (AM - MA) - \mathbf{1}\mu^T = \sum_{i=1}^n w_{ii}u_iu_i^T + \sum_{i,j\in[n]:i\neq j}\langle u_i, \mathbf{1}\rangle w_ju_iu_j^T - \mathbf{1}\mu^T$$

$$= \sum_{i=1}^n w_{ii}u_iu_i^T + \sum_{i,j\in[n]:i\neq j}\langle u_i, \mathbf{1}\rangle w_ju_iu_j^T - \sum_{i=1}^n\langle u_i, \mathbf{1}\rangle u_i\mu^T$$

$$= \sum_{i=1}^n w_{ii}u_iu_i^T - \sum_{i=1}^n\langle u_i, \mathbf{1}\rangle w_iu_iu_i^T$$

$$= -\sum_{i\in[n]:|\langle u_i, \mathbf{1}\rangle|\leq n^{-\epsilon/16}}u_iu_i^T - C\sum_{i\in[n]:|\langle u_i, \mathbf{1}\rangle|\geq n^{-\epsilon/16}}u_iu_i^T \overset{\triangle}{=} D.$$

Now, we show that $\|D\|_F^2 = o(n)$ asserting that $R - (AM - MA) - \mathbf{1}\mu^T$ is small. As the Frobenius norm is unitary invariant, we have

$$\|D\|_F = \sqrt{\# \left\{ |\langle u_i, \mathbf{1} \rangle| \leq n^{-\epsilon/16} \right\} + C^2 \# \left\{ |\langle u_i, \mathbf{1} \rangle| \geq n^{-\epsilon/16} \right\}}$$

$$\leq \sqrt{\# \left\{ |\langle u_i, \mathbf{1} \rangle| \leq n^{-\epsilon/16} \right\} + C^2 n}.$$

Next, we get a handle on $C$ using the following claim, proved at the end of this section.

**Claim 9.** *There exists a constant $c > 0$ such that, for large enough $n > 0$, we have, with a probability of at least $1 - e^{-cn^{1-\epsilon/16}}$, $\# \left\{ |\langle u_i, \mathbf{1} \rangle| \leq n^{-\epsilon/16} \right\} \leq 3n^{1-\epsilon/16}$.*

To prove the above claim, we use the fact that the orthonormal eigenvector basis of a GOE matrix is uniformly distributed on $\mathcal{S}_{n-1}$ and the details are provided in Appendix C.2. Thus, $\langle u_i, \mathbf{1} \rangle$ is approximately a standard normal and so $\# \left\{ |\langle u_i, \mathbf{1} \rangle| \leq n^{-\epsilon/16} \right\} \approx n^{1-\epsilon/16}$. By the above claim

$$0 \geq C = 1 - \frac{n}{n - \# \left\{ |\langle u_i, \mathbf{1} \rangle| \leq n^{-\epsilon/16} \right\}} \geq 1 - \frac{n}{n - 3n^{1-\epsilon/16}} \geq -5n^{-\epsilon/16}, \tag{9}$$

where the last inequality follows for $n > 0$ large enough (depending on $\epsilon$). The bound on $C$ implies

$$\|D\|_F \leq \sqrt{\# \left\{ |\langle u_i, \mathbf{1} \rangle| \leq n^{-\epsilon/16} \right\} + C^2 n} \leq \sqrt{3n^{1-\epsilon/16} + 25n^{1-\epsilon/8}} \leq 2n^{1/2-\epsilon/32}, \tag{10}$$

where the last inequality holds for $n$ large enough (depending on $\epsilon$). Now, for any $X \in \mathcal{B}_n$, we have

$$\langle R - (AM - MA) - \mathbf{1}\mu^T, X \rangle = \langle R, X \rangle - \langle AM - MA, X \rangle - \langle \mathbf{1}\mu^T, X \rangle$$

$$= \sum_{i,j \in [n]: i \neq j} X_{ij} - \langle M, AX - XA \rangle - \langle \mu, \mathbf{1} \rangle = \sum_{i,j \in [n]: i \neq j} X_{ij} - \langle M, AX - XA \rangle,$$

where the last inequality holds as $\langle \mu, \mathbf{1} \rangle = 0$ by construction. Thus, we have

$$\sum_{i,j \in [n]: i \neq j} X_{ij} = \langle M, AX - XA \rangle + \langle X, D \rangle \overset{(10)}{\leq} \langle M, AX - XA \rangle + 2n^{1-\epsilon/32},$$

where the last inequality holds as $\langle X, D \rangle \leq \|X\|_F \|D\|_F$ and $\|X\|_F^2 \leq n \max_{j \in [n]} \sum_{i=1}^n X_{ij}^2 \leq n \max_{j \in [n]} \sum_{i=1}^n |X_{ij}| = n$ for any $X \in \mathcal{B}_n$. Now, to complete the proof, we upper bound $\langle M, AX - XA \rangle$ below. Let $X = \sum_{i,j=1}^n x_{ij} u_i u_j^T$ for some $x_{ij} \in \mathbb{R}$ for all $i, j \in [n]$. Then,

$$\langle M, AX - XA \rangle = \left\langle \sum_{i,j \in [n]: i \neq j} \frac{\tilde{w}_{ij}}{\lambda_i - \lambda_j} u_i u_j^T, \sum_{i,j \in [n]: i \neq j} x_{ij}(\lambda_i - \lambda_j) u_i u_j^T \right\rangle$$

$$= \left\langle \sum_{i,j \in [n]: i \neq j} \frac{\tilde{w}_{ij}}{|\lambda_i - \lambda_j| + n^{-1-\epsilon}} u_i u_j^T, \sum_{i,j \in [n]: i \neq j} x_{ij} \left( |\lambda_i - \lambda_j| + n^{-1-\epsilon} \right) u_i u_j^T \right\rangle$$

$$\overset{*}{\leq} \sqrt{\sum_{i,j \in [n]: i \neq j} \frac{\tilde{w}_{ij}^2}{(|\lambda_i - \lambda_j| + n^{-1-\epsilon})^2}} \cdot \left( \|AX - XA\|_F + n^{-0.5-\epsilon} \right) \tag{11}$$

where $(*)$ follows from the Cauchy-Schwarz inequality. In addition, we upper bound the second term by using the triangle inequality and noting that $n \geq \|X\|_F^2 = \sum_{i,j=1}^n x_{ij}^2 \geq \sum_{i,j \in [n]: i \neq j} x_{ij}^2$ for all $X \in \mathcal{B}_n$. In addition, we also note that $\sum_{i,j \in [n]: i \neq j} x_{ij}^2 (\lambda_i - \lambda_j)^2 = \|AX - XA\|_F^2$. Now, we focus on getting a handle on the term involving the eigenvalue separation $(\lambda_i - \lambda_j)$. We have

$$\sqrt{\sum_{i,j \in [n]: i \neq j} \frac{\tilde{w}_{ij}^2}{(|\lambda_i - \lambda_j| + n^{-1-\epsilon})^2}}$$

$$= (1 - C) \sqrt{\sum_{i,j \in [n]: i \neq j} \frac{\langle u_i, \mathbf{1} \rangle^2}{\langle u_j, \mathbf{1} \rangle^2 (|\lambda_i - \lambda_j| + n^{-1-\epsilon})^2} \mathbb{1} \left\{ |\langle u_j, \mathbf{1} \rangle| \geq n^{-\epsilon/16} \right\}}$$

$$\overset{*}{\leq} 2n^{\epsilon/16} \sqrt{\sum_{i,j \in [n]: i \neq j} \frac{\langle u_i, \mathbf{1} \rangle^2}{(|\lambda_i - \lambda_j| + n^{-1-\epsilon})^2}} \overset{**}{\leq} 2n^{\epsilon/8} \sqrt{\sum_{i,j \in [n]: i \neq j} \frac{1}{(|\lambda_i - \lambda_j| + n^{-1-\epsilon})^2}}, \tag{12}$$

where $(*)$ follows for $n$ large enough by (9). In addition, $(**)$ follows with probability $1 - ne^{-n^{\epsilon/8}/8} - ne^{-n^{1/4}}$. Indeed, one can combine the union bound over $i \in [n]$ with $|\langle u_i, \mathbf{1} \rangle| = \frac{|\langle z, \mathbf{1} \rangle|}{\|z\|_2} \leq \frac{2|\langle z, \mathbf{1} \rangle|}{\sqrt{n}} \leq n^{\epsilon/16}$, where $z \sim N(0, I_n)$. The second inequality holds with probability $1 - e^{-n^{1/4}}$ for $n$ large enough (e.g. see: [11, Lemma 15]). In addition, the last inequality holds with probability $1 - e^{-n^{\epsilon/8}/8}$ (e.g. see: [11, Lemma 13]). Now, we get a handle on the eigenvalue separation in the following claim:

**Claim 10.** *For $n$ large enough, w.p. $1 - o_n(1)$, we have $\sum_{i,j \in [n]: i \neq j} \frac{1}{(|\lambda_i - \lambda_j| + n^{-1-\epsilon})^2} \leq n^{3+3\epsilon/2}$.*

Note that naively bounding $(|\lambda_i - \lambda_j| + n^{-1-\epsilon})^{-2} \leq n^{2+2\epsilon}$ results in a weaker upper bound of $n^{4+2\epsilon}$. We perform a much tighter analysis by carefully using the tail bounds for the separation of the eigenvalue from [29]. In particular, depending on whether $|i - j|$ is small or large, we perform a separate analysis to lower bound $|\lambda_i - \lambda_j|$. The analysis is especially delicate when $|i - j|$ is small: directly using the tail bounds of [29] combined with a union bound over $i \in [n]$ is not sufficient. We instead use the tail bounds to first compute $\mathbb{E}\left[ \sum_{|i-j|=O(1)} \frac{1}{(|\lambda_i - \lambda_j| + n^{-1-\epsilon})^2} \right]$ and then use the Markov's inequality to get a tighter upper bound. Note that the trick to introduce $n^{-1-\epsilon}$ in (11) is crucial here to ensure the finiteness of the expectation. The proof details of this claim are deferred to Appendix C.3. Now, we continue with the proof of Lemma 6 below.

Using the above claim along with (11) and (12), for $n$ large enough, we get

$$\langle M, AX - XA \rangle \leq 2n^{3/2+7\epsilon/8} \left( \|AX - XA\|_F + n^{-0.5-\epsilon} \right)$$

which further implies

$$\sum_{i \neq j} X_{ij} \leq 2n^{3/2+7\epsilon/8} \|AX - XA\|_F + 2n^{1-\epsilon/8} + 2n^{1-\epsilon/32}$$

$$\leq 2n^{3/2+7\epsilon/8} \|AX - XA\|_F + 4n^{1-\epsilon/32},$$

where the last inequality holds for $n$ large enough (depending on $\epsilon$). This completes the proof with probability $1 - o_n(1)$ for $n$ large enough by the union bound over all the bounds in the proof. $\quad\square$

## 5    Simulations

We conduct simulations on the Gaussian Wigner Model to verify our results and provide further insights (The code is publicly available at `https://github.com/smv30/convex_rel_for_graph_alignment`). We set $n = 400$ (unless otherwise specified) and consider $\sigma \in \{0, 0.1, 0.2, \ldots, 1\}$. For these parameters, we test the performance of three convex relaxations: GRAMPA [11], simplex [2], and Birkhoff. We set the regularization parameter to $0.2$ in GRAMPA as suggested by the authors. The convex relaxations are solved using the `cvxpy` library in Python using SCS (Splitting Conic Solver) and we set `use_indirect` to `True`, recommended for large instances for better memory management. We then project the solution of the convex relaxation to the set of permutation matrices using the Hungarian algorithm. For each $n$ and $\sigma$, we repeat the simulation 10 times and report the average fraction of correctly matched vertices in Figure 1. We run the simulations on a CPU with 50GB of memory and impose a maximum run time of 3 hours for each instance.

On the left of Figure 1, we observe that the Birkhoff relaxation outperforms both Simplex and GRAMPA by exactly aligning all vertices for $\sigma$ up to $0.5$. On the other hand, the performance of Simplex and GRAMPA starts degrading for $\sigma = 0.4$ and $\sigma = 0.2$ respectively. Such a strong empirical performance motivates the theoretical analysis of the Birkhoff relaxation.

In the center plot of Figure 1, we plot the fraction of matched vertices and $\|X^\star - \Pi^\star\|_F / \sqrt{n}$ as a function of $\sigma$ for Birkhoff relaxation. In conjunction with Theorem 1 (case I), $\|X^\star - \Pi^\star\|_F / \sqrt{n}$ increases rapidly and converges to one as a function of $\sigma$. However, even when $\|X^\star - \Pi^\star\|_F / \sqrt{n}$ is close to one for $\sigma \in [0.3, 0.5]$, the Birkhoff relaxation still manages to align all the vertices. In particular, while the optimal solution $X^\star$ is not close to $\Pi^\star$, it has a slight bias toward $\Pi^\star$ as opposed to other permutation matrices. So, $X^\star$ is projected to $\Pi^\star$ in the post-processing step of projection onto the set of permutation matrices. This result suggests that Birkhoff relaxation combined with post-processing could succeed beyond $\sigma \sim n^{-0.5}$.

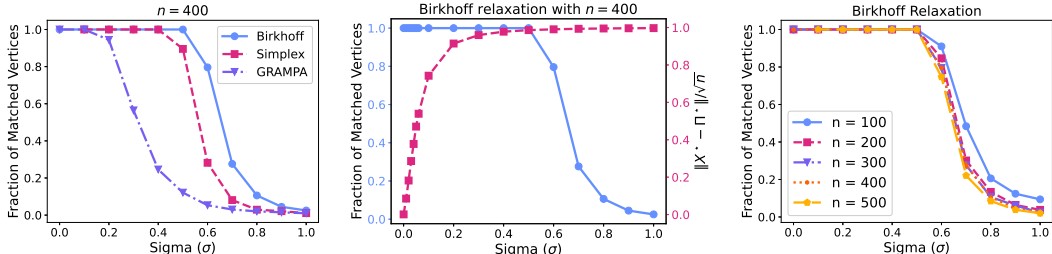

Figure 1: Fraction of Matched Vertices and $\|X^\star - \Pi^\star\|_F/\sqrt{n}$ as a function of $\sigma$ for the Gaussian Wigner Model: Performance of GRAMPA [11], Simplex [2], and Birkhoff Relaxations

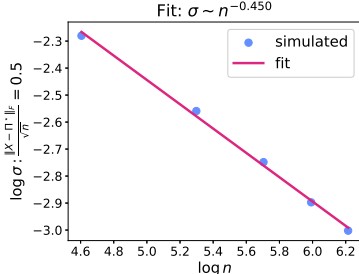

Figure 2: Log-log plot for $\sigma$ such that $\frac{\|X^\star - \Pi^\star\|_F}{\sqrt{n}} = 0.5$ as a function of $n$.

In the right plot of Figure 1, we test the performance of Birkhoff relaxation for $n \in \{100, 200, 300, 400, 500\}$. Although the fraction of correctly matched vertices as a function of $\sigma$ worsens as $n$ increases, the performance degradation is gradual. Such an empirical observation hints that the Birkhoff relaxation combined with post-processing could succeed for nearly constant $\sigma$. Although we take a first step in this direction, non-trivial ideas are needed for stronger guarantees.

In Figure 2, we empirically compute the value of $\sigma$ at which $\|X^\star - \Pi^\star\|_F$ transitions from $o(\sqrt{n})$ to $\Omega(\sqrt{n})$. In particular, for each $n \in \{100, \ldots, 500\}$, we implement the Birkhoff relaxation for $\sigma \in \{0, 0.01, \ldots, 1\}$ to infer the value of $\sigma$ at which $\|X^\star - \Pi^\star\|_F/\sqrt{n} = 0.5$. A linear regression between these thresholds $(\log \sigma)$ and $\log n$ outputs a slope of $-0.45$, supporting the phase transition at $\sigma = \Theta(1/\sqrt{n})$ as in Theorem 1.

# 6 Conclusion and Future Work

In this paper, we study Birkhoff relaxation, a tight convex relaxation of the Quadratic Assignment Problem (QAP). The input is sampled from a Gaussian Wigner model with correlation $1/\sqrt{1 + \sigma^2}$. We show that $X^\star$ (optimal solution of Birkhoff relaxation) is a small perturbation of $\Pi^\star$ (optimal solution of QAP) when $\sigma \ll n^{-1}$, and $X^\star$ is far away from $\Pi^\star$ when $\sigma \gg n^{-0.5}$. More specifically, we show that $\|X^\star - \Pi^\star\|_F^2 = o(n)$ when $\sigma = o(n^{-1})$ and $\|X^\star - \Pi^\star\|_F^2 = \Omega(n)$ when $\sigma = \Omega(n^{-0.5})$. This result allows us to align $1 - o(1)$ fraction of vertices whenever $\sigma = o(n^{-1})$.

Based on heuristic calculations using the population version of the Birkhoff relaxation, we believe the condition to ensure $\|X^\star - \Pi^\star\|_F^2 = o(n)$ can be improved to $\sigma = o(n^{-0.5})$, which is an immediate future work. In addition, while we show that $\|X^\star - \Pi^\star\|_F^2 = \Omega(n)$ when $\sigma = \Omega(n^{-0.5})$, it does not imply the failure of Birkhoff relaxation, i.e., one can still post-process $X^\star$ appropriately to recover $\Pi^\star$. Hence, another future direction is to establish the success of post-processing procedures combined with the Birkhoff relaxation for $\sigma$ greater than $n^{-0.5}$.

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

## A    Rounding Procedures

*Proof of Corollary 2.* We bound the number of incorrectly matched indices as follows:

$$\sum_{i=1}^{n} \mathbb{1}\left\{\hat{\pi}(i) \neq \pi^{\star}(i)\right\} \leq \sum_{i=1}^{n} \mathbb{1}\left\{X_{i\pi^{\star}(i)}^{\star} \leq \frac{1}{2}\right\} \leq 4\sum_{i=1}^{n}\left(1 - X_{i\pi^{\star}(i)}^{\star}\right)^{2}$$

$$\leq 4\|X^{\star} - \tilde{\Pi}^{\star}\|_{F}^{2} \overset{*}{=} 4\|X^{\star} - \Pi^{\star}\|_{F}^{2} \leq 40n^{1-\epsilon/32},$$

where $(*)$ holds as $\tilde{\Pi} = \Pi^{\star}$ with probability $1 - o(1)$ by [15, 32]. The last inequality is followed by the second part of Theorem 1 with probability $1 - o(1)$. □

*Proof of Corollary 3.* Similar to the proof of Theorem 1, let $\tilde{\Pi}^{\star} = \Pi^{\star} = I$ without loss of generality. Now, let $\tilde{\Pi} \in \arg\max_{\Pi \in \mathcal{P}_{n}}\langle X^{\star}, \Pi\rangle$. Then, we have

$$\langle X^{\star}, \tilde{\Pi}\rangle \geq \langle X^{\star}, I\rangle \overset{(8)}{\geq} n - o(n).$$

Also, we have

$$\langle X^{\star}, \tilde{\Pi}\rangle = \langle I, \tilde{\Pi}\rangle + \langle X^{\star} - I, \tilde{\Pi}\rangle \leq \langle I, \tilde{\Pi}\rangle + \|X^{\star} - I\|_{F}\|\tilde{\Pi}\|_{F} \leq \langle I, \tilde{\Pi}\rangle + o(n),$$

where the last inequality follows by Theorem 1 for $\sigma \geq n^{-1-\epsilon}$. Combining the above two inequalities, we get $\langle I, \tilde{\Pi}\rangle \geq n - o(n)$ showing that the Hungarian solution $\tilde{\Pi}$ overlaps with the correct permutation $I$ for at least $1 - o(1)$ fraction of the vertices, concluding the proof. □

## B    Technical Details for Part I: Well-Separation

### B.1    Proof of Claim 4

We have

$$\|AX^{\star} - X^{\star}B\|_{F}^{2} \leq \frac{1}{n^{2}}\|AJ - JB\|_{F}^{2} = \frac{1}{n^{2}}\|AJ - JA - \sigma JZ\|_{F}^{2}$$

$$\overset{*}{\leq} \frac{3}{n^{2}}\left(\|AJ\|_{F}^{2} + \|JA\|_{F}^{2} + \sigma^{2}\|JZ\|_{F}^{2}\right)$$

$$\overset{**}{\leq} \frac{6}{n}\sum_{i=1}^{n}\left(\sum_{k=1}^{n}A_{ik}\right)^{2} + \frac{3}{n}\sum_{i=1}^{n}\left(\sum_{k=1}^{n}Z_{ik}\right)^{2} \leq 9n^{\epsilon}, \tag{13}$$

where $(*)$ uses the inequality $(a + b + c)^{2} \leq 3(a^{2} + b^{2} + c^{2})$, and $(**)$ holds as $\sigma \leq 1$. Moreover, the last inequality follows with probability at least $1 - 4ne^{-n^{\epsilon}/4}$ for $n$ large enough. Indeed, $\sum_{k=1}^{n}A_{ik} \sim N(0, 1 + 1/n)$ and so $\left|\sum_{k=1}^{n}A_{ik}\right| \leq n^{\epsilon/2}$ for all $i \in [n]$ with probability $1 - 2e^{-n^{\epsilon}/4}$ for $n$ large enough. Thus, since $A, Z$ are i.i.d., by the union bound, $\sum_{i=1}^{n}\left(\sum_{k=1}^{n}A_{ik}\right)^{2} \leq n^{1+\epsilon}$ and $\sum_{i=1}^{n}\left(\sum_{k=1}^{n}Z_{ik}\right)^{2} \leq n^{1+\epsilon}$ with probability at least $1 - 4ne^{-n^{\epsilon}/4}$.

### B.2    Proof of Claim 5

For any $i, j \in [n]$, we have

$$(AZ)_{ij} = \sum_{k=1}^{n}A_{ik}Z_{kj} \overset{*}{\leq} \frac{2n^{\epsilon/2}}{\sqrt{n}}\|A_{i,:}\|_{2} \quad \textit{w.p. at least } 1 - e^{-n^{\epsilon}}$$

$$\overset{**}{\leq} \frac{4n^{\epsilon/2}}{\sqrt{n}},$$

where $(*)$ is true because $\{Z_{kj} : k \in [n]\}$ are independent from each other, and independent from $A$. Thus, $(*)$ follows by conditioning on $A$ and using [11, Lemma 13]. Next, $(**)$ holds with probability at least $1 - e^{-\sqrt{n}}$ as $\|A_{i,:}\|_{2} \leq \frac{\sqrt{2}}{\sqrt{n}}\|z\|_{2}$, where $z \sim N(0, I_{n})$ and by [11, Lemma 15],

$\|z\|_2^2 \le n + \tilde{c}\sqrt{n} \le 2n$ for sufficiently large $n$. Thus, by the union bound, $(AZ)_{ij} \le 4n^{\epsilon/2-1/2}$ with probability at least $1 - 2e^{-n^\epsilon}$. By further employing union bound on $i, j \in [n]$, with probability at least $1 - 2n^2 e^{-n^\epsilon}$, we have

$$\max_{i \neq j} |(AZ - ZA)_{ij}| \le 2\max_{i \neq j} |(AZ)_{ij}| \le \frac{8n^{\epsilon/2}}{\sqrt{n}}.$$

## C  Technical Details for Part II: Small-Perturbation

### C.1  Proof of Lemma 7

We prove the desired inequality in the event where

$$\|Z\|_F^2 \le \tilde{c}^2 n,$$

for some constant $\tilde{c} > 0$, which occurs with probability at least $1 - e^{-n}$ if $\tilde{c}$ is large enough, from [11, Lemma 15], because $\|Z\|_F^2$ is a sum of $\frac{n(n+1)}{2}$ independent squared Gaussian variables with variance $\frac{2}{n}$.

First, we show that $\|AX^\star - X^\star B\|_F = O(\sigma\sqrt{n})$. As $X^\star$ is the minimizer of (2) and $I \in \mathcal{B}_n$,

$$\|AX^\star - X^\star B\|_F^2 \le \|A - B\|_F^2 = \sigma^2 \|Z\|_F^2 \le \tilde{c}^2 \sigma^2 n.$$

We now obtain an upper bound on $\|AX^\star - X^\star A\|_F$ using the upper bound on $\|AX^\star - X^\star B\|_F$:

$$\begin{aligned}
\|AX^* - X^*A\|_F &\le \|AX^* - X^*B\|_F + \|X^*(A - B)\|_F \\
&= \|AX^* - X^*B\|_F + \sigma\|X^*Z\|_F \\
&\le \tilde{c}\sigma\sqrt{n} + \sigma\|X^*\|_2\|Z\|_F \\
&\le \tilde{c}\sigma\sqrt{n} + \tilde{c}\sigma\sqrt{n}\|X^*\|_2 \\
&\le 2\tilde{c}\sigma\sqrt{n}.
\end{aligned}$$

The last inequality follows as $X^\star \in \mathcal{B}_n$ and the spectral norm of a doubly stochastic matrix is at most 1, e.g., see [20]. It completes the proof with $c = 2\tilde{c}$.

### C.2  Proof of Claim 9

Let $U \in \mathbb{R}^{n \times n}$ be the matrix whose lines are $u_1, \ldots, u_n$. It is orthogonal, and its distribution is uniformed on the set of orthogonal matrices. Therefore, $U\mathbf{1}$ is uniformly distributed in $\sqrt{n}\mathcal{S}_{n-1}$, so that it is equal in distribution to $\sqrt{n}\frac{z}{\|z\|_2}$, for $z \sim N(0, I_n)$. In particular,

$$\begin{aligned}
&\mathbb{P}\left(\#\left\{|\langle u_i, \mathbf{1}\rangle| \le n^{-\epsilon/16}\right\} > 3n^{1-\epsilon/16}\right) \\
&= \mathbb{P}\left(\#\left\{|z_i| \le n^{-1/2-\epsilon/16}\|z\|_2\right\} > 3n^{1-\epsilon/16}\right) \\
&\le \mathbb{P}\left(\#\left\{|z_i| \le 2n^{-\epsilon/16}\right\} > 3n^{1-\epsilon/16}\right) + \mathbb{P}\left(\|z\|_2 > 2\sqrt{n}\right) \\
&\overset{*}{\le} \mathbb{P}\left(\sum_{i=1}^n \mathbb{1}\left\{|z_i| \le 2n^{-\epsilon/16}\right\} > 3n^{1-\epsilon/16}\right) + e^{-c_1 n} \quad \text{(for some constant } c_1 > 0) \\
&\overset{**}{\le} \mathbb{P}\left(\sum_{i=1}^n \left(\mathbb{1}\left\{|z_i| \le 2n^{-\epsilon/16}\right\} - \mathbb{E}\mathbb{1}\left\{|z_i| \le 2n^{-\epsilon/16}\right\}\right) > n^{1-\epsilon/16}\right) + e^{-c_1 n} \\
&\overset{***}{\le} e^{-c_2 n^{1-\epsilon/16}} + e^{-c_1 n} \quad \text{(for some constant } c_2 > 0).
\end{aligned}$$

Inequality $(*)$ is true, again, from [11, Lemma 15]. Inequality $(**)$ is true because

$$\mathbb{E}\left(\sum_{i=1}^n \mathbb{1}\left\{|z_i| \le 2n^{-\epsilon/16}\right\}\right) = \frac{n}{\sqrt{2\pi}}\int_{-2n^{-\epsilon/16}}^{2n^{-\epsilon/16}} e^{-\frac{t^2}{2}}\,dt \le 2n\sqrt{\frac{2}{\pi}}n^{-\epsilon/16} \le 2n^{1-\epsilon/16}.$$

Inequality $(***)$ is due to Bernstein's inequality. This completes the proof of claim with $c = c_2/2$ as $e^{-c_2 n^{1-\epsilon/16}} + e^{-c_1 n} \le e^{-c_2 n^{1-\epsilon/16}/2}$ for $n$ large enough.

## C.3 Proof of Claim 10

We pick $L > 0$ (constant depending on $\epsilon$) such that $\min_{|i-j|>L} |\lambda_i - \lambda_j| \geq n^{-1-\epsilon/2}$ with probability at least $1 - o(1)$ by [29, Corollary 2.5]. So we have

$$
\sum_{i \neq j} \frac{1}{(|\lambda_i - \lambda_j| + n^{-1-\epsilon})^2} = \sum_{k=0}^{\left\lceil \frac{n}{L} \right\rceil} \sum_{|i-j| \in [kL, (k+1)L]} \frac{1}{(|\lambda_i - \lambda_j| + n^{-1-\epsilon})^2}
$$

$$
\leq \sum_{|i-j| \leq L} \frac{1}{(|\lambda_i - \lambda_j| + n^{-1-\epsilon})^2} + \sum_{k=1}^{\left\lceil \frac{n}{L} \right\rceil} \sum_{|i-j| \in [kL, (k+1)L]} \frac{1}{(\lambda_i - \lambda_j)^2}
$$

$$
\leq \sum_{|i-j| \leq L} \frac{1}{(|\lambda_i - \lambda_j| + n^{-1-\epsilon})^2} + \sum_{k=1}^{\left\lceil \frac{n}{L} \right\rceil} \sum_{|i-j| \in [kL, (k+1)L]} \frac{n^{2+\epsilon}}{k^2}
$$

$$
\leq \sum_{|i-j| \leq L} \frac{1}{(|\lambda_i - \lambda_j| + n^{-1-\epsilon})^2} + n^{3+\epsilon} L \sum_{k=1}^{\left\lceil \frac{n}{L} \right\rceil} \frac{1}{k^2}
$$

$$
\leq \sum_{|i-j| \leq L} \frac{1}{(|\lambda_i - \lambda_j| + n^{-1-\epsilon})^2} + 10 L n^{3+\epsilon}.
$$

Now, we bound the remaining terms below. We have

$$
\mathbb{E}\left[ \frac{1}{(|\lambda_j - \lambda_i| + n^{-1-\epsilon})^2} \right]
$$

$$
= \int_0^\infty \mathbb{P}\left( \frac{1}{(|\lambda_j - \lambda_i| + n^{-1-\epsilon})^2} > x \right) dx
$$

$$
= \int_0^\infty \mathbb{P}\left( |\lambda_j - \lambda_i| < \frac{1}{\sqrt{x}} - n^{-1-\epsilon} \right) dx
$$

$$
= \int_0^{n^{2+2\epsilon}} \mathbb{P}\left( |\lambda_j - \lambda_i| < \frac{1}{\sqrt{x}} - n^{-1-\epsilon} \right) dx + \int_{n^{2+2\epsilon}}^\infty \mathbb{P}\left( |\lambda_j - \lambda_i| < \frac{1}{\sqrt{x}} - n^{-1-\epsilon} \right) dx
$$

$$
\leq \int_0^{n^{2+2\epsilon}} \mathbb{P}\left( |\lambda_j - \lambda_i| < \frac{1}{\sqrt{x}} \right) dx \leq 2 c_0 n^{2+\epsilon},
$$

where the last inequality holds by [29, Corollary 2.2] for some constant $c_0 > 0$. Using the above inequality, we get

$$
\mathbb{E}\left[ \sum_{|i-j| \leq L} \frac{1}{(|\lambda_j - \lambda_i| + n^{-1-\epsilon})^2} \right] \leq 2 c_0 L n^{3+\epsilon}.
$$

Now, by the Markov's inequality, with probability $1 - n^{-\epsilon/4}$, we get

$$
\sum_{|i-j| \leq L} \frac{1}{(|\lambda_j - \lambda_i| + n^{-1-\epsilon})^2} \leq 2 c_0 L n^{3+5\epsilon/4}.
$$

Thus, we get

$$
\sum_{i \neq j} \frac{1}{(|\lambda_i - \lambda_j| + n^{-1-\epsilon})^2} \leq 2 c_0 L n^{3+5\epsilon/4} + 10 L n^{3+\epsilon} \leq n^{3+3\epsilon/2},
$$

where the last inequality holds for $n$ large enough (depending on $\epsilon$). The proof is now complete.

