# OpenReview forum: "Graph Alignment via Birkhoff Relaxation"
_NeurIPS.cc/2025/Conference — NeurIPS 2025 poster_

### Official Review · Reviewer_59z5 · 2025-06-24

**Clarity:** 3
**Significance:** 3
**Originality:** 3
**Rating:** 5
**Confidence:** 3

**Summary:**

The paper "Graph Alignment via Birkhoff Relaxation" analyzes the Birkhof relaxation of quadratic assignment problems (QAPs) from a theoretical perspective. It considers the setting of inputs following a Gaussian Wigner Model, and gives guarantees based on the dependence of the standard deviation $\sigma$ of the perturbation (in the input) on the dimensionality of the problem. The analysis results in a state-of-the-art condition for the provable success of the Birkhoff relaxation. It also indicates a phase transition for standard deviations that decay slower. A brief numerical study indicates that the success of the relaxation might go beyond the provable guarantee and that post-processing rounding schemes seem to yield strong results even in the case where the result  of the relaxation itself is provably well-separated from the true permutation.

**Questions:**

Please consider improving the readability of the manuscript for a broader audience. In particular, please address the explicit suggestions for an improved clarity listed under "Strength and Weaknesses".

In the checklist, the authors state that "Our code is a simple Python script to solve convex optimization problems using cvxpy. Such an approach is ubiquitous, so we decided not to make the code open-source. However, we are happy to make it open-source if the reviewers deem it to be essential."  Despite the simplicity of the experiments, I strongly recommend making the source code publically available. This allows other researchers to reproduce that paper's results exactly with minimal effort.

**Ethical Concerns:**

["NO or VERY MINOR ethics concerns only"]

**Final Justification:**

While I agree with my fellow reviewers that the practical impact could be stronger (including dedicated solvers for competitive runtimes), I think it is a nice theoretical contribution. Moreover, the authors promised to release their code. I will keep my initial rating.

**Limitations:**

yes

**Paper Formatting Concerns:**

no concern

**Quality:**

3

**Strengths And Weaknesses:**

The paper addresses an interesting and important problem. To the best of my knowledge, it makes an important theoretical contribution, advancing the provable guarantees in the form of error estimates for the Birkhoff relaxation of the quadratic assignment problem under specific assumptions on the input matrices. Although the numerical results do not necesarily indicate that the error bounds (or conditions) are tight, I consider this work to be an important step towards a better understanding of the Birkhoff relaxation.

In terms of the presentation, the paper is not very easy to follow. This is partly due to the theoretical, mathematically challenging nature of the topic. Yet, moving some more techincal aspects into the appendix to generate space for a little more context to address a broader audience could be helpful (e.g. possibly introducing the Birkhoff polytope formaly). Some specific aspects that could improve the presentation:
- I would have found it helpful if alternative relaxation methods were briefly introduced. For instance, I could imagine that "simplex" (reference [1]) is a weaker relaxation obtained by dropping the sum-to-one constraint along the columns (or rows), but currently understanding the relaxation requires looking at reference [1].
- The variable $J$ (equation after l. 110) is not defined but used many times throughout the manuscript. Its meaning can be deduced from the context but this is not friendly for a quick read.
- In line 50, I think it should read $B= A+\sigma Z$ (not $B^{\pi*}$), and the definition of $B^{\pi^*}$ follows from $B$ in the next line.
- After reading in line 123 that the analysis in [1] is restricted to $\sigma =0$, and that the work presented here allows $\sigma>0$, I was surprised to read "we consider a simpler optimization problem corresponding to $\sigma=0$ in (2)" in line 157. It subsequently becomes clear that this is just an intermediate step, but informing the reader about the big picture of the proof strategy would be beneficial.
- The bound $\sqrt{n}\geq ||X||_F^2$ is a typo (but it is used correctly, without the square, in the proof).
- I'd recommend to consider different colors for the middle plot of Fig. 1, or insert a title "Birkhoff Relaxation for n=400". As the left plot has the same colors and a legend that does not apply to the middle plot, this can be misleading when having a quick look only.
- Typo in line 260, "left plot" should be "right plot".

---

> ### Author Rebuttal · Authors · 2025-07-28
>
> **Comment:** Please consider improving the readability of the manuscript for a broader audience. In particular, please address the explicit suggestions for an improved clarity listed under "Strength and Weaknesses".
>
> **Response:** Thank you so much for carefully reading our manuscript, suggesting several expositional improvements, and catching some typos! We are grateful for your time and efforts. We will implement all the suggested changes in the manuscript.
>
> **Comment:** In the checklist, the authors state that "Our code is a simple Python script to solve convex optimization problems using cvxpy. Such an approach is ubiquitous, so we decided not to make the code open-source. However, we are happy to make it open-source if the reviewers deem it to be essential." Despite the simplicity of the experiments, I strongly recommend making the source code publically available. This allows other researchers to reproduce that paper's results exactly with minimal effort.
>
> **Response:** Thank you so much for this comment. We will make the code open source and include it with the paper.

---

### Official Review · Reviewer_dUSb · 2025-06-28

**Clarity:** 4
**Significance:** 3
**Originality:** 3
**Rating:** 4
**Confidence:** 3

**Summary:**

This paper studies the problem of graph alignment in the Gaussian Wigner Model. The goal is to align the vertex sets of two correlated matrices A, B where B has an unknown permutation applied to it. The authors consider the Birkhoff relaxation of the problem, which is a well-known (but poorly studied) relaxation of the QAP formulation. When the correlation between A and B is $o(n^{-1})$, the Frobenius norm between the optimal solution and the true permutation matrix is $o(n)$. In turn, this guarantees that a simple rounding technique recovers the true permutation. On the other hand, when the correlation between A and B is $\Omega(n^{-0.5}),$ then the Frobenius norm between the optimal solution and the true permutation matrix is $\Omega(n)$. However, the latter result does not rule out the possibility of rounding the relaxed solution.

**Questions:**

In Equation (3), don’t we need an upper bound on $\Vert Z^2 \Vert_F$ in order to derive a lower bound on $\Vert I – X^{\star} \Vert_F$?

I am not following the equality in Line 198. Could you please clarify?

**Ethical Concerns:**

["NO or VERY MINOR ethics concerns only"]

**Final Justification:**

Viewed as a theoretical contribution, I support the acceptance of this paper. In their rebuttals, the authors proposed adding several concrete elements to the paper, which I view positively.

**Limitations:**

yes

**Quality:**

3

**Strengths And Weaknesses:**

Analyzing Birkhoff relaxation has been a difficulty in the graph alignment literature, so I view this paper as making a solid step forward, even though the correlation algorithmic guarantees are weaker than those of competing approaches (GRAMPA, simplex relaxation). Still, empirical results show that the Birkhoff relaxation strategy (+ Hungarian algorithm) performs better than those approaches.

The cited works are incomplete. Most crucially, reference [8] is cited but not included in Section 1.3. Also, a recent paper of Gaudio, Racz, and Sridhar (“Average-case and smoothed analysis of graph isomorphism”) provides improved guarantees for matching in the noiseless Erdos-Renyi setting.

Line 215: “the triangle’s inequality”

Line 260: “left plot” should be “right plot”

Corollary 2 uses the greedy approach for matching. However, the simulations use the Hungarian algorithm. Therefore, I don’t see the simulations as validating Theorem 1, but rather as a standalone investigation. This shortcoming should be acknowledged, and there should be an empirical evaluation that uses the greedy approach.

---

> ### Author Rebuttal · Authors · 2025-07-28
>
> **Comment:** In Equation (3), don’t we need an upper bound on $\Vert Z^2 \Vert_F$ in order to derive a lower bound on $\Vert I – X^{\star} \Vert_F$?
>
> **Response:** You are correct that we need an upper bound on $\Vert Z^2 \Vert_F$. We obtain such an upper bound as follows: $\Vert Z^2 \Vert_F \leq \sqrt{n}\Vert Z^2 \Vert_2 \leq \sqrt{n}\Vert Z \Vert_2^2 \leq c\sqrt{n}$ for some $c>0$ large enough with high probability. The first two inequalities hold for any square matrix. The third inequality holds as $Z$ is GOE and so its spectral norm is bounded by a constant (say 3). These steps are mentioned in Line 148 of the paper.
>
> **Comment:** I am not following the equality in Line 198. Could you please clarify?
>
> **Response:** Thank you for your comment. Define a diagonal matrix $\tilde{D}$ with $\tilde{D}(ii) = -1$ for all $i \in [n]$ such that $|\langle \mathbf{1}, u_i \rangle| \leq n^{-\epsilon/16}$ and $\tilde{D}(ii)=-C$ otherwise. Also, let $U$ to be the unitary matrix with columns $\{u_i\}_{i=1}^n$. Then, we have $D = U \tilde{D} U^T$. As the Frobenius norm is unitary invariant, we have $\|D\|_F = \|U \tilde{D} U^T\|_F = \|\tilde{D}\|_F$, and so the result follows. We will mention that we are using unitary invariance of the Frobenius norm to improve the exposition, thank you!
>
> **Comment:** Corollary 2 uses the greedy approach for matching. However, the simulations use the Hungarian algorithm. Therefore, I don’t see the simulations as validating Theorem 1, but rather as a standalone investigation. This shortcoming should be acknowledged, and there should be an empirical evaluation that uses the greedy approach.
>
> **Response:** Thank you for raising this interesting point! You are correct to point out that Corollary 2 deals with a simple rounding procedure and not the Hungarian algorithm. Actually, Theorem 1 (Eq. (6) more precisely) also implies the success of the Hungarian rounding procedure. The following simple proof suffice: Let $\Pi^\star =I$ be the correct permutation (WLOG), $X^\star$ be the solution of Birkhoff relaxation and $\tilde{\Pi} \in \arg\max_{\Pi \in \mathcal{P}_n} \langle X^\star, \Pi\rangle $. Then, we have $\langle X^\star, \tilde{\Pi}\rangle \geq \langle X^\star, I\rangle \geq n - o(n)$ by Eq. (6). Also, we have $\langle X^\star, \tilde{\Pi}\rangle = \langle I, \tilde{\Pi}\rangle + \langle X^\star-I, \tilde{\Pi}\rangle \leq \langle I, \tilde{\Pi}\rangle + \|X^\star-I\|_F, \|\tilde{\Pi}\|_F \leq \langle I, \tilde{\Pi}\rangle + o(n)$ by Theorem 1. Combining the two inequalities above, we get $\langle I, \tilde{\Pi} \rangle \geq n - o(n)$, showing that the Hungarian solution $\tilde{\Pi}$ overlaps with the correct permutation $I$ for at least $1-o(1)$ fraction of the vertices. We will include this result as another corollary in the paper, thank you for raising this point!

---

> > ### Comment · Reviewer_dUSb · 2025-08-04
> >
> > Thank you for addressing my questions! I felt that the authors also answered the questions of the other reviewers comprehensively. Experts on graph alignment view understanding the Birkhoff relaxation as a valuable goal, so I feel this paper makes an important theoretical contribution.

---

### Official Review · Reviewer_gpnQ · 2025-06-30

**Clarity:** 4
**Significance:** 2
**Originality:** 3
**Rating:** 5
**Confidence:** 3

**Summary:**

The paper considers the graph-alignment (quadratic assignment) problem under the Gaussian Wigner Model and delivers the first guarantees for the Birkhoff relaxation. It shows a sharp phase transition: when noise is $o(1/n)$, the relaxed solution remains close to the true permutation, while for noise on $\Omega(1/\sqrt{n})$ it drifts significantly away. The author's analysis relies on a dual certificate that respects the non-negativity constraints of the Birkhoff polytope, combined with sensitivity analysis of convex problems. The experiments are simulated on graphs of up to 500 nodes to verify theoretical results.

**Questions:**

1. The authors should further discuss the comparison with previous works and the generalizability of theoretical frameworks.

2. Would the authors consider sampling $\sigma$ more densely within the intermediate range $[1/n,1/\sqrt{n})]$ and reporting how the empirical error aligns with the predicted behavior?

3. The reported runtime (up to 3 hours) appears significantly higher than related works on graph matching [1, 2], where solving times are typically reported in seconds. Could the authors explain this difference?

4. The authors mention that the code is simple and thus not released. However, reproducibility is critical, especially for optimization problems where solver settings can influence results.

[1]. Bernard, Florian, Daniel Cremers, and Johan Thunberg. "Sparse quadratic optimisation over the stiefel manifold with application to permutation synchronisation." Advances in Neural Information Processing Systems 34 (2021): 25256-25266.

[2]. Dröge, Hannah, et al. "Kissing to find a match: efficient low-rank permutation representation." Advances in Neural Information Processing Systems 36 (2023): 48459-48471.

**Ethical Concerns:**

["NO or VERY MINOR ethics concerns only"]

**Final Justification:**

This rebuttal address most of my concerns, and I decide to raise my rating.

**Limitations:**

The authors partially address the limitations of their work in Sections 2 and 6. However, a more explicit discussion of the two main limitations mentioned above would strengthen the paper.

**Paper Formatting Concerns:**

The subscript “F” in the Frobenius norm should be set in an upright font.

**Quality:**

3

**Strengths And Weaknesses:**

## Strengths
1. The paper provides the first formal guarantee for the Birkhoff relaxation in the graph alignment problems and identifies a meaningful threshold on the noise level that separates success from failure.

2. The paper is well written and easy to follow. The main contributions are presented in a straightforward manner, and the core theoretical result is introduced early on, accompanied by helpful intuition.

## Weaknesses
1. The relative improvement over prior work is difficult to assess. Although the paper offers a comprehensive literature review, it does not quantitatively compare its theoretical results with existing works. A concise comparison table—summarizing assumptions, noise thresholds, and relaxation types—would help clarify the novelty of this work.

2. The analysis is limited to the correlated Gaussian Wigner model. It is unclear whether the proposed proof techniques can generalize to new settings.

3. While the theory predicts a sharp transition based on the noise level, this phenomenon is not fully explored in the experiments. In particular, it would be valuable to examine how the error evolves more finely across the noise regime, especially when $\sigma$ lies between $1/n$ and $1/\sqrt{n}$.

---

> ### Author Rebuttal · Authors · 2025-07-28
>
> **Comment:** The relative improvement over prior work is difficult to assess. Although the paper offers a comprehensive literature review, it does not quantitatively compare its theoretical results with existing works. A concise comparison table—summarizing assumptions, noise thresholds, and relaxation types—would help clarify the novelty of this work.
>
> **Response:** Adding a concise comparison table to compare our results with the literature is a good idea. We will add such a table to the paper.
>
> Table headers: Paper, Algorithm Type, Algorithm Name, Noise Threshold
>
> Row 1: [Ganassali-Lelarge-Massoulie 2022], Top Eigenvector Alignment, EIG1,  $\sigma=\Theta(n^{-7/6})$
>
> Row 2: [Fan-Mao-Wu-Xu 2022], Regularized Convex Relaxation, GRAMPA, $\sigma = O(1/\log n)$
>
> Row 3: [Araya-Tyagi 2023], Simplex Relaxation, Simplex, $\sigma=0$
>
> Row 4: Our work, Birkhoff Relaxation, Birkhoff, $\sigma = O(n^{-1})$
>
> Caption: Comparison of the spectral and optimization-based algorithms for graph alignment on the Gaussian Wigner Model.
>
> Discussion: Our established threshold of $\sigma = O(n^{-1})$ improves over the guarantees in [Ganassali-Lelarge-Massoulie 2022] [Araya-Tyagi 2023]. On the other hand, GRAMPA has better theoretical guarantees; however, the Birkhoff relaxation has better empirical performance, hence the need to develop a rigorous analysis for this strategy as well.
>
> [Ganassali-Lelarge-Massoulie 2022] Ganassali, Luca, Marc Lelarge, and Laurent Massoulié. "Spectral alignment of correlated Gaussian matrices." Advances in Applied Probability 54.1 (2022): 279-310.
>
> [Fan-Mao-Wu-Xu 2022] Fan, Z., Mao, C., Wu, Y. et al. Spectral Graph Matching and Regularized Quadratic Relaxations I Algorithm and Gaussian Analysis. Found Comput Math 23, 1511–1565 (2023)
>
> [Araya-Tyagi 2023] Araya, Ernesto, and Hemant Tyagi. "Graph Matching via convex relaxation to the simplex." Foundations of Data Science 7.2 (2023): 464-501.
>
> **Comment:** The analysis is limited to the correlated Gaussian Wigner model. It is unclear whether the proposed proof techniques can generalize to new settings.
>
> It is an interesting research direction to extend these results to more general input distributions. We discuss below the technical challenges that entail these generalizations:
>
> **Extensions of our results:** It would be desirable to consider the input matrices sampled from a certain generic distribution with some concentration properties, e.g., subgaussian and subexponential.
>
> **Well Separation Result:** We believe this part of the theorem can be generalized to more general symmetric matrices $A$ with i.id. entries that concentrate sufficiently, e.g., subgaussian concentration would suffice. In particular, we only need the following concentration properties of our random matrices $A, Z$: $\Vert Z\Vert_2 \leq c$, $\Vert Z\Vert_F^2 \geq n/2$, $\\max\_{i,j \\in [n]} |(AZ)\_{ij}| \leq 2n\^{\epsilon/2-0.5}$, $\\max_{i \\in [n]}\\big|\\sum\_{k=1}\^n A\_{ik}\\big| \\leq n\^{\epsilon/2}$. We will add that as a remark in the paper.
>
> **Small Perturbation Result:** The main difficulties are to get a handle on the eigenvalues and eigenvectors of $A$. Below, we outline the instances in the proof where we explicitly use the properties of a GOE matrix:
>
> *Eigenvalue separation [Claim 8]:* We use tail bounds on the eigenvalues of a random matrix from [5]. The results of [5] are applicable for all Wigner matrices (i.id. subgaussian entries), and so Claim 8 can be extended to Wigner matrices.
>
> *Eigenvector Concentration [Claim 7]:* To prove Claim 7, we rely on the fact that the orthonormal eigenvectors of a GOE matrix is uniformly distributed on $\mathcal{S}_{n-1}$ which guarantees that $\langle \mathbf{1}, u_i\rangle$ has the same distribution as $z/\|z\|_2$, where $z \sim N(0, I_n)$. Thus, we can get upper and lower concentrations on $|\langle \mathbf{1}, u_i \rangle|$. This step is the bottleneck, as we require such concentration results for general Wigner matrices. We will discuss this difficulty in the paper to highlight the limitations.
>
> **Comment:** Would the authors consider sampling $\sigma$ more densely within the intermediate range $[1/n,1/\sqrt{n})]$ and reporting how the empirical error aligns with the predicted behavior?
>
> **Response:** That is an excellent suggestion! We simulated the Birkhoff relaxation for several values of $\sigma \in [0, 0.1]$ to estimate the value of $\sigma$ for which $\|X^\star-\Pi^\star\|_F/\sqrt{n} = 0.5$. We obtain the following values of $\sigma$ as the output: 0.105 (for $n=100$); 0.077 (for $n=200$); 0.063 (for $n=300$); 0.054 (for $n=400$); 0.050 (for $n=500$). A linear regression between these thresholds ($\log \sigma$) and $\log n$ outputs a slope of $-0.475$, verifying the phase transition at $\sigma = \Theta(n^{-0.5})$. We will add these simulation results to the paper.
>
> **Comment:** The reported runtime (up to 3 hours) appears significantly higher than related works on graph matching [1, 2], where solving times are typically reported in seconds. Could the authors explain this difference?
>
> **Response:** Thank you so much for this comment. As the contributions of this paper are the theoretical analysis, and the Birkhoff relaxation is well-known in the literature, we did not attempt to improve the run-times and simply employed the SCS solver using the cvxpy library of Python. The previous work [Araya Tyagi 2023] focuses on the computational times of these convex relaxations and reports a run-time of 35 seconds for the Birkhoff relaxation with $n=300$ using the method of Alternating
> Direction Method of Multipliers (ADMM). The SCS solver, on the other hand, requires about 15 mins to converge for $n=300$, suggesting that alternative optimization techniques can perform substantially better computationally. As our focus is on the theoretical analysis, we did not attempt to improve the run-time. As a sanity check, note that our empirical results (Fig. 1 left) closely match those of [Fig. 1a, 1].
>
> [Araya Tyagi 2023] Araya, Ernesto, and Hemant Tyagi. "Graph Matching via convex relaxation to the simplex." Foundations of Data Science 7.2 (2023): 464-501.
>
> **Comment:** The authors mention that the code is simple and thus not released. However, reproducibility is critical, especially for optimization problems where solver settings can influence results.
>
> **Response:** Thank you so much for this comment. We will make the code open source and include it with the paper.

---

> > ### Comment · Reviewer_gpnQ · 2025-08-03
> >
> > Thank you for the thorough response, especially the insightful discussion regarding the method's extensibility. I will raise my rating.
> > To clarify, given the theoretical nature of the paper, I do not believe it is necessary for the authors to optimize the runtime. However, as the method is heavily theory-driven, I believe the authors—being most familiar with its design—are best positioned to improve its efficiency to a more practical level. Such enhancements may be difficult for others to achieve without their direct involvement.

---

### Official Review · Reviewer_a4Lo · 2025-07-08

**Clarity:** 3
**Significance:** 1
**Originality:** 4
**Rating:** 4
**Confidence:** 3

**Summary:**

This paper presents a theoretical analysis of Birkhoff relaxation for graph alignment under the Gaussian Wigner Model (claimed as a first). The authors establish theoretical guarantees showing that when $\sigma = o(n^{-1})$, the optimal solution $X^* $ approximates the true permutation with $||X^* - \Pi^*||^2_F = o(n)$, while solutions become well-separated when $\sigma = \Omega(n^{-\frac{1}{2}})$. The work employs dual certificate construction techniques to handle non-negativity constraints and provides rigorous mathematical analysis including eigenvalue separations and concentration inequalities. The paper provides a synthetic simulation showcasing its prowess over the traditional baseline.

**Questions:**

1. **Real-world relevance:** Is it a valid question to ask how the noise levels in your theoretical model relate to those encountered in practical applications mentioned in the introduction? What range of σ values is typical in practice? Are applications grouped based on where such a relaxation can be applied and where it cannot be?

2. **Scope extension:** Can these techniques extend beyond the Gaussian Wigner model to more general graph alignment settings? What are the main technical challenges when you are thinking of, let us say, sub-Gaussian, Bernoulli, etc.? Is there practical relevance to these other extension models?

**Ethical Concerns:**

["NO or VERY MINOR ethics concerns only"]

**Final Justification:**

I am updating my score as I found the reponse from the authors partially satisfactory

**Limitations:**

It would be great if the authors added a dedicated limitations section discussing the points touched upon in the Weaknesses and Questions section.

**Paper Formatting Concerns:**

No major formatting issues were observed. The paper follows standard mathematical formatting conventions with proper theorem environments, notation, and references.

**Quality:**

3

**Strengths And Weaknesses:**

**Strengths:**
- **Novel theoretical contribution:** A rigorous theoretical analysis of Birkhoff relaxation for graph alignment, which serves as a significant contribution to support the method's empirical success.
- **Technical rigor:** Mathematical analysis is thorough with dual certification and proofs demonstrate rigorous treatment (eigenvalue separations, concentration inequalities, etc).
- **Clear phase transition:** Establishes a meaningful theoretical phase regime where alignment succeeds versus fails.

**Weaknesses:**
While I see that the paper seems like a nice theoretical contribution, I still think that the paper reads as something developed in a vacuum. More discussion of why these results are an important event, which is needed for the next step in a practically relevant application, would be a great addition. There is limited discussion on the impact of this particular result proved in the paper.

- **Limited practical relevance:** The paper fails to connect theoretical insights to downhill ramification (practical algorithm design, impact on real-world applications, etc.).
- **Narrow experimental validation:** Evaluation restricted to synthetic Gaussian Wigner experiments. The paper is missing evaluations on standard graph matching benchmarks or real-world datasets. Graph alignment has extensive applications in computational biology, social networks, and computer vision; however, there are no specific ablation studies on the impact of specific graph strucutre on the bounds, even empirically would be great to see. Helps talk about future directions.

---

> ### Author Rebuttal · Authors · 2025-07-28
>
> **Comment: Real-world relevance:** Is it a valid question to ask how the noise levels in your theoretical model relate to those encountered in practical applications mentioned in the introduction? What range of $\sigma$ values is typical in practice? Are applications grouped based on where such a relaxation can be applied and where it cannot be?
>
> **Response:** Thank you for your comment. We more broadly discuss the implications of our result for practice.
>
> **Real-world relevance:** Doubly stochastic relaxations for graph matching problems has been an attractive approach in practice, with strong empirical performance for shape matching [Aflalo-Bronstein-Kimmel 2015] and ordering images in a grid [Nadav-Maron-Lipman 2017]. It is also observed in [Lyzinski et.al. 2015] that Birkhoff relaxation, when combined with indefinite relaxation, yields excellent empirical performance on real datasets. While these results highlight the usefulness of the Birkhoff relaxation to practice due to its attractive computational and empirical performance, there is limited understanding of its theoretical performance. Our paper is one of the first results in filling this gap in the literature.
>
> **Analysis of Graph Matching:** Graph Matching is an instance of the quadratic assignment problem, which is NP-hard in the worst case. Thus, one cannot guarantee the success of convex relaxations (e.g., Birkhoff) in the worst case. A ubiquitous approach in the literature is then to consider a stylized model, where the inputs are sampled from certain distributions. We take this approach where the problem instance is sampled as a correlated Gaussian Wigner Model with correlation $1/\sqrt{1+\sigma^2}$. Imposing such distributional assumptions on the input graphs allows for mathematical analysis.
>
> **Role of $\sigma$:** Note that $\sigma=0$ is the special case of the graph isomorphism problem. As the value of $\sigma$ increases, the correlation between the two graphs reduces, which makes it harder to align them. In other words, noise increases, making it harder to extract the correct signal. Thus, $\sigma$ serves as a tuning parameter to increase the hardness of the problem. The takeaway is then to establish that the Birkhoff relaxation succeeds for large values of $\sigma$, establishing the robustness of the relaxation. We make progress in this direction by improving upon the previously known result [Araya-Tyagi 2023] that shows the simplex relaxation succeeds for $\sigma=0$. These results contribute to explaining the strong empirical performance of the Birkhoff relaxation.
>
>
> [Aflalo-Bronstein-Kimmel 2015] Aflalo, Yonathan, Alexander Bronstein, and Ron Kimmel. "On convex relaxation of graph isomorphism." Proceedings of the National Academy of Sciences 112.10 (2015): 2942-2947
>
> [Nadav-Maron-Lipman 2017] Nadav, D. Y. M., Haggai Maron, and Yaron Lipman. "DS++: A flexible, scalable and provably tight relaxation for matching problems." ACM Transactions on Graphics 36.6 (2017): a184.
>
> [Lyzinski et.al. 2015] Lyzinski, Vince, et al. "Graph matching: Relax at your own risk." IEEE transactions on pattern analysis and machine intelligence 38.1 (2015): 60-73.
>
> [Araya-Tyagi 2023] Araya, Ernesto, and Hemant Tyagi. "Graph Matching via convex relaxation to the simplex." Foundations of Data Science 7.2 (2023): 464-501.
>
>
>
> **Comment: Scope extension:** Can these techniques extend beyond the Gaussian Wigner model to more general graph alignment settings? What are the main technical challenges when you are thinking of, let us say, sub-Gaussian, Bernoulli, etc.? Is there practical relevance to these other extension models?
>
> Thank you for this insightful question! It is an interesting research direction to extend these results to more general input distributions. We discuss below the technical challenges that entail these generalizations:
>
> **Extensions of our results:** It would be desirable to consider the input matrices sampled from a certain generic distribution with some concentration properties, e.g., subgaussian and subexponential.
>
> **Well Separation Result:** We believe this part of the theorem can be generalized to more general symmetric matrices $A$ with i.id. entries that concentrate sufficiently, e.g., subgaussian concentration would suffice. In particular, we only need the following concentration properties of our random matrices $A, Z$: $\Vert Z\Vert_2 \leq c$, $\Vert Z\Vert_F^2 \geq n/2$, $\\max\_{i,j \\in [n]} |(AZ)\_{ij}| \\leq 2n\^{\\epsilon/2-0.5}$, $\max_{i \in [n]}\big|\sum_{k=1}^n A_{ik}\big| \leq n^{\epsilon/2}$. We will add that as a remark in the paper.
>
>
> **Small Perturbation Result:** The main difficulties are to get a handle on the eigenvalues and eigenvectors of $A$. Below, we outline the instances in the proof where we explicitly use the properties of a GOE matrix:
>
> *Eigenvalue separation [Claim 8]:* We use tail bounds on the eigenvalues of a random matrix from [5]. The results of [5] are applicable for all Wigner matrices (i.id. subgaussian entries), and so Claim 8 can be extended to Wigner matrices.
>
> *Eigenvector Concentration [Claim 7]:* To prove Claim 7, we rely on the fact that the orthonormal eigenvectors of a GOE matrix is uniformly distributed on $\mathcal{S}_{n-1}$ which guarantees that $\langle \mathbf{1}, u_i\rangle$ has the same distribution as $z/\|z\|_2$, where $z \sim N(0, I_n)$. Thus, we can get upper and lower concentrations on $|\langle \mathbf{1}, u_i \rangle|$. This step is the bottleneck, as we require such concentration results for general Wigner matrices. We will discuss this difficulty in the paper to highlight the limitations.

---

### Decision · Program_Chairs · 2025-09-17

**Decision:**

Accept (poster)

**Comment:**

All reviewers found that the paper contains solid theoretical contribution that advances our understanding of the Birkhoff relaxation of the graph alignment problem. As the authors revise their paper, they are urged to give a more thorough discussion of the practical significance of their work, as well as a detailed comparison of the results obtained in this work with those in the literature.